# *Saprochaete clavata* Infection in Immunosuppressed Patients: Systematic Review of Cases and Report of the First Oral Manifestation, Focusing on Differential Diagnosis

**DOI:** 10.3390/ijerph18052385

**Published:** 2021-03-01

**Authors:** Carlo Lajolo, Cosimo Rupe, Anna Schiavelli, Gioele Gioco, Elisabetta Metafuni, Maria Contaldo, Simona Sica

**Affiliations:** 1Head and Neck Department, “Fondazione Policlinico Universitario A. Gemelli—IRCCS”, School of Dentistry, Università Cattolica del Sacro Cuore, Largo A. Gemelli, 8, 00168 Rome, Italy; carlo.lajolo@policlinicogemelli.it (C.L.); anna.schiavelli@gmail.com (A.S.); gioele.gioco@gmail.com (G.G.); 2Dipartimento di Diagnostica per Immagini, Radioterapia Oncologica ed Ematologia, Fondazione Policlinico Universitario A. Gemelli IRCCS, 00168 Rome, Italy; bettymetafuni@yahoo.it (E.M.); simona.sica@policlinicogemelli.it (S.S.); 3Sezione di Ematologia, Dipartimento di Scienze Radiologiche ed Ematologiche, Università Cattolica del Sacro Cuore, 00168 Rome, Italy; 4Multidisciplinary Department of Medical-Surgical and Dental Specialties, University of Campania Luigi Vanvitelli, Via Luigi de Crecchio, 6, 80138 Naples, Italy; maria.contaldo@unicampania.it

**Keywords:** *Saprochaete clavata*, *Geotrichum clavatum*, rare mycoses

## Abstract

**Background:** Saprochaete clavata infection is an emerging issue in immunosuppressed patients, causing fulminant fungaemia. The purpose of this systematic review of cases is to retrieve all cases of *S. clavata* infection and describe oral lesions as the first manifestation of *S. clavata* infection. **Methods:** We report the first case of intraoral *S. clavata* infection in Acute Myeloid Leukemia (AML) affected subject, presenting as multiple grayish rapidly growing ulcerated swellings, and provide a review of all published cases of infection caused by *S. clavata*, according to PRISMA (Preferred Reporting Items for Systematic Reviews and Meta-Analyses) guidelines, conducted by searching SCOPUS, Medline, and CENTRAL databases. Only articles in English were considered. Individual patient data were analyzed to identify risk factors for *S. clavata* infection. **Results:** Seventeen of 68 retrieved articles were included in the review reporting data on 96 patients (mean age 51.8 years, 57 males and 38 females). Most cases were disseminated (86) with a 60.2% mortality rate. Ninety-five were hematological patients, with AML being the most common (57 cases). **Conclusions:**
*S. clavata* infection in immunosuppressed patients has a poor prognosis: middle-age patients, male gender and Acute Myeloid Leukemia should be considered risk factors. In immunosuppressed patients, the clinical presentation can be particularly unusual, imposing difficult differential diagnosis, as in the reported case.

## 1. Introduction

Opportunistic fungal infections represent a major cause of morbidity and mortality in patients with malignant hematological diseases [1,2,3]. Aspergillus and Candida species are the most frequent etiological agents, while Geotrichum genus has only recently been detected as an emerging opportunistic pathogen in hematological patients [4,5].

Geotrichum is a ubiquitous, filamentous, yeast-like genus of fungi composed of 18 species, among which *Geotrichum clavatum* (reclassified as *Saprochaete clavata*) and *Geotrichum capitatum* (reclassified as *Saprochaete capitata*) are the most common pathogens for human infection [6].

The genus Geotrichum can be found worldwide in soil, water, air, wood, animals, and dairy products. Moreover, it has also been isolated as a commensal fungus in the respiratory secretions and gastrointestinal tract of Mediterranean subjects [7,8]. Orofecal transmission is the most common source of infection [9]; in particular, ingestion of cheese seems to be the most common way, as these fungi are used in cheese maturing.

When a *S. clavata* infection arises in immunosuppressed patients, it may cause fulminant fungaemia with multi-organ involvement and it has a 60% to 80% mortality rate. Disseminated infection initially arises with nonspecific symptoms (i.e., fever, diarrhea, and pulmonary symptoms), and most infections are mainly diagnosed by blood culture [10]. The clinical presentation in immunocompetent patients is less severe and rarely reported [11].

The most commonly reported risk factors associated with *S. clavata* infection include hematological malignancy, prolonged neutropenia, high-dose corticosteroid therapy, broad-spectrum antibiotic therapy, previous gastrointestinal colonization, and central venous catheters [4,12,13,14].

The purpose of this systematic review of cases is to report all cases of *S. clavata* infection and to describe the first case of oral lesions as the first manifestation of *S. clavata* infection in an immunosuppressed subject, especially focusing on the differential diagnosis of the oral lesions.

## 2. Methods

The present systematic review was conducted according to the PRISMA (Preferred Reporting Items for Systematic Reviews and Meta-Analyses) guidelines. A comprehensive and systematic electronic search in Medline (via PubMed), Scopus, and the Cochrane Central Register of Controlled Trials (CENTRAL) from database inception to December 2020 was conducted. The date of the last search was 16 December 2020. The database search was conducted by using a combination of the following MeSH terms and free text words: “Saprochaete” OR “Geotrichum” AND “rare mycoses”. A supplementary manual search was conducted for articles published from the journals’ inception dates and December 2020 in the following journals: Oral Oncology; Clinical Oral Investigations; Journal of Oral Pathology and Medicine; Oral Surgery, Oral Medicine, Oral Pathology, and Oral Radiology; Head and Face Medicine; and Oral Diseases. In addition, the bibliographies of all the selected articles were reviewed. Inclusion criteria were: full papers; English literature; observational clinical studies, namely, case reports, perspective and retrospective (cohort and case-control) and randomized clinical trials (RCTs); patients who received an *S. clavata* infection. All of the articles reporting cases of *S. clavata* infection were included in this review. No exclusion criteria were applied. Two reviewers (C.R. and C.L.) conducted the screening, independently and in duplicate, by using specially designed data extraction forms. For each included article, the variables collected included the year of publication, country, number of patients, gender and age of patients, underlying disease, clinical manifestation of the infection, and the final outcome of patient. Any disagreements were resolved by a third reviewer (A.S.). All data collected were analyzed using a computer program (SPSS v 21.0, Chicago, IL, USA).

## 3. Results

The initial search of the electronic databases yielded 68 articles, and the manual search yielded 2 additional articles. In total, 46 articles were removed on the basis of the review of their titles and abstracts, reporting other fungal infections; thus, 22 full-text articles were selected. Of these studies, 17 were included in the review (Figure 1).

Pooled data from the literature review, stratified according to Geographic Distribution, Age, Gender, Anamnesis, and Clinical Features of patients, are shown in Table 1. This systematic review of the literature retrieved 96 cases of *S. clavata* infection. Most of the cases have been diagnosed in France (46 cases) or Italy (35 cases) and middle age (51.8 years) and male gender (57 males and 38 females) seem to be more frequently affected. Most cases were disseminated (86), leading to death (60.2% of mortality rate). Other sites involved in the infection were gastrointestinal (24 patients had diarrhea) and pulmonary (27). Only one case arose with a single organ involvement (a splenic abscess). Considering the underlying immunodepression cause, 95 were hematological patients (comprising our case report) and only 1 was affected by polycystic kidney disease (PCKD). Among the hematological diseases, Acute Myeloid Leukemia was the most common (57 cases), whereas Lymphomas or Acute Lymphoid Leukemia were diagnosed in 12 and 10 cases, respectively. As concerning infection treatment, data about 48 patients were available; most of the patients (36) were treated by Azoles (30 patients by Voriconazole, 4 by Posaconazole). Other treatments were mainly based on Amphotericin B (27 patients), often in association with Azoles.

## 4. Case Report

A 56-year-old Caucasian man affected by acute myeloid leukemia (AML) was admitted to the Hematological Department of Policlinico A. Gemelli (Rome, Italy) for febrile neutropenia with multiple oral swellings. The patient’s medical history revealed previous diagnoses of myelodysplastic syndrome (2016), refractory anemia with excess blasts type 1 (RAEB-1), which was treated with ten cycles of decitabine and an allogenic hematopoietic stem cell transplantation. This recurring disease evolved into Acute Myeloid Leukemia (2018). Consequently, re-induction chemotherapy was administered using Chlorambucil and Cytarabine first, followed by a combination of Azacitidine and Venetoclax. In October 2019, the patient was hospitalized for febrile neutropenia with severe thrombocytopenia. At that moment, the patient was receiving antifungal prophylaxis with Posaconazole 300 mg/die. Blood culture was performed and the research for yeasts was negative. The physical examination did not retrieve any skin lesion. The oral examination revealed that three rapidly growing asymptomatic ulcerated swellings, located on both the buccal mucosa and the left mandible, arose 15 days before (Figure 2, Figure 3 and Figure 4).

Lesions were light-grayish in color, friable, with a 4 cm diameter, and protruded from the mucosa, interfering with chewing. After a platelet transfusion and local anesthesia (Carbocaine 2% with epinephrine 1:100,000; Dentsply, Verona, Italy), an incisional biopsy was performed together with a microbiological sampling for fungi. The pathology revealed an inflammatory process in association with purulent necrosis and microbial colonies; no granulomas were detected in the specimen. Part of the sample was discharged in liquid Amies medium, then streak-plated on Sabouraud dextrose agar plates supplemented with gentamicin and chloramphenicol (Bio-Rad, Hercules, CA, USA) and BBL CHROMagar Candida plate (BD). We identified species using Bruker Biotyper version MBT 3.1 matrix-assisted laser desorption/ionization time-of-flight (MALDI-TOF) mass spectrometry (Bruker) and nucleotide sequence analysis of the internal transcribed spacer (ITS) regions of the rRNA gene. Microbiological sampling was thus positive for *S. clavata*, thus the final diagnosis indicated an intraoral fungal infection due to *S. clavata*. The Minimal inhibitory concentrantions (MICs) of antifungal drugs were determined in parallel according to the European Committee on Antimicrobial Susceptibility Testing (EUCAST) standardized broth microdilution method [27].

Given the results of the antifungal susceptibility testing, a therapy based on Voriconazole 200 mg bid was set and resulted in the resolution of the oral lesions and improvement of the overall condition. Two weeks after the beginning of the therapy, the patient was dismissed, continuing the therapy with Voriconazole for two months. Nevertheless, in February 2020, the patient died from a worsening of his general conditions and underlying disease persistence.

## 5. Discussion

Deep fungal infections represent a major cause of morbidity and mortality in onco-hematological patients undergoing immunosuppressing chemotherapies or those who have undergone bone marrow transplantation. In fact, fungi are responsible for approximately 20% of microbiologically documented intensive care unit (ICU) infections. In the last decade, the incidence of invasive fungal infections has steadily increased as a result of the increasing numbers of both immunocompromised and critically ill patients [1,2,3,4,10,11,28].

In spite of this, the life expectancy of onco-hematological patients is rising thanks to improved diagnostic and therapeutical techniques; on the other side, the onco-hematological population is increasingly susceptible to rare infections, like *S. clavata*, which has been recognized as an emerging issue [5]. Three major sources of infection have been recognized: contaminated medical devices or dishes or dairy products have been advocated as major sources of *S. clavata* [9], but, considering published literature, the source of infection was unclear in the majority of the cases [26,28].

Even if the literature review identified only case reports and case series, probably due to the newly diagnosed entity, the articles included in this systematic review showed some results (Table 2) that should be taken into consideration for clinicians and future studies.

*S. clavata* infection seems to be more common in males (57 to 38 patients), even if more diagnosed cases are needed to state that male gender is a risk factor for *S. clavata* infection. The mean age of affected patients was 51.8 years. Middle age should be considered as a risk factor for the development of this infection, even if our results showed how even younger or pediatric patients could be affected. In order to better assess the risk factors linked to this infection, clinical studies providing a higher level of evidence should be performed.

According to the available literature, the majority of patients affected by *S. clavata* had received a diagnosis of Acute Myeloid Leukemia and underwent a therapy based on Cytarabine, an antineoplastic drug that can alter digestive mucosa. It has been hypothesized that the mucosal alteration caused by Cytarabine allows *S. clavata* translocation from the gastrointestinal tract, thus promoting the diffusion and the infection [9]. Although this hypothesis could explain the pathogenetic mechanism favoring *S. clavata* infection, it is not clear why mainly hematological patients are affected compared to other immunosuppressed patients (95 to 1 in the reported cases) [29,30].

Although *S. clavata* infection usually arises with nonspecific symptoms, it causes disseminated infection in almost all of the cases (86 out of 96 of the included cases), threatening the survival of infected patients. In this light, it is crucial to reach an early diagnosis in order to start the correct treatment.

Considering the treatment, three major classes of antifungal agents can be considered: polyenes, azoles, and echinocandins. Polyenes (i.e., amphotericin B, nystatin) bind to ergosterol of fungal membrane, causing its disruption (fungicidal); Azoles (i.e., Voriconazole, Itraconazole) act by inhibiting ergosterol synthesis in the endoplasmic reticulum of the fungal cell (fungistatic) and echinocandins (i.e., Caspofungin, Anidulafungin, Micafungin) and inhibit the synthesis of some components of the fungi cell wall. According to the available literature, *S. clavata* seems to be intrinsically resistant to echinocandins [31], whereas Voriconazole, Itraconazole, Posaconazole, Amphotericin B, and 5-fluorocytosine showed some efficacy [9,32]; Voriconazole appears to be the most suitable treatment since it has shown effectiveness both in vitro and in vivo [3,9,14,33].

The results of this systematic review show how Voriconazole and Amphotericin B are the most used treatment: 5 patients were treated by Voriconazole alone, all of them survived. Two patients were treated by Amphotericin B alone, but their outcome was not successful. The association between these two drugs was the treatment of choice for the majority of patients (21): eleven of them survived, while 10 patients died. Although only scattered data could be obtained from our literature review, the hypothesis that Voriconazole is the most effective treatment can be supported, even if further studies are needed to confirm it.

Considering our experience, even if our patient was already, at the admission, undergoing a prophylactic treatment based on Posaconazole (300 mg die) to prevent opportunistic infections, the susceptibility testing confirmed that both Posaconazole and Voriconazole were effective drugs against *S. clavata,* but oral lesions healed only after the use of Voriconazole (200 mg bid): this strengthened the hypothesis that Voriconazole is the most effective treatment.

This case report describes the first case of oral lesions as the first manifestation of *S. clavata* infection and the literature review did not find any other case with oral involvement, thus differential diagnosis was particularly demanding. The infection manifested as three rapidly growing asymptomatic ulcerated swellings, located on both the buccal mucosa and the left mandible, that arose 15 days before. Lesions were light grayish in color, friable, with a 4 cm diameter, and protruded from the mucosa, interfering with chewing (Figure 1). Since clinical presentation of common and rare diseases can be really unusual in immunosuppressed patients, the diagnosis of these lesions was particularly challenging, but some clinical characteristics drove the differential diagnosis: rapid growth, ulcerating feature, bilateral onset, and absence of any other symptom (i.e., pain, itching, burning sensation). For these reasons, oral or deep fungal infections, other opportunistic infections, relapses, or new onset of hematological diseases and granulomatous diseases have been considered in the differential diagnosis (Table 3).

Candida and Aspergillus are the most common opportunistic fungal infections affecting the oral cavity [5]. In this case, oral candidiasis was excluded because its most common clinical presentations (Pseudomembranous Candidiasis and Acute erythematous candidiasis) do not comprise rapidly growing asymptomatic ulcerated swellings; nevertheless, a microbiological culture was performed to exclude C. *albicans* and other species infection or its superimposed infection.

Aspergillosis is a fungal disease characterized by noninvasive and invasive forms; in immunocompromised hosts, the invasive form can cause disseminated and life-threatening infections. This opportunistic fungal infection is reported as the second most prevalent one worldwide [50]. Aspergillosis spores can colonize the brain, the bones, the lungs, or the endocardium [5,51,52,53,54]. Signs and symptoms that may affect the head and neck district include sinusitis, oropharyngeal colonization, pain, swelling, ulceration, necrosis, and palatal perforation. In addition, the hyphae are able to penetrate the oral mucosa and the arterial wall, which may lead to hematogenous spread, thrombosis, or infarction [55,56]. Considering the clinical presentation of our case, oral Aspergillosis lesions arise generally as swelling with a necrotic ulcerated base, classically located on the palate or posterior tongue [5]. Nevertheless, microbiological culture on the biopsy sample was performed to exclude the diagnosis of oral Aspergillosis.

Histoplasmosis is another clinical entity that can appear as a rapidly growing asymptomatic oral ulceration. Although it is the most common fungal infection in the United States, Histoplasmosis in Europe is rare [57]. Almost 95% of the immunocompetent patients that are exposed to the Histoplasma capsulatum never develop symptoms [58]. Yet, inhaling a large number of spores can cause symptoms even among healthy patients. Histoplasmosis has two manifestations: acute or chronic/disseminated. Only this latter form can affect the oral mucosa, causing painful chronic ulcerations, single or multiple, nonhealing and indurated [59]. The hypothesis of a case of Histoplasmosis was rejected according to the medical history of the patient, thanks to microbiological culture and pathology.

Among other infections, Syphilis, Tuberculosis, and Actinomycosis must be considered in the differential diagnosis. Treponema Pallidum infection in immunosuppressed individuals can cause diffused ulcerations of the oral cavity, with severe clinical appearance. Nevertheless, in our case, the TPHA (Treponema Pallidum Hemoagglutination Assay) was negative at admission in the hospital [60,61]. Tuberculosis causes ulcerations as well, affecting mainly the tongue and the palate, and usually have an “infiltrating” feature more than exophytic [62]; furthermore, pathology can reveal typical caseous tuberculous granulomas which can drive the diagnosis.

Actinomycosis is a chronic bacterial disease that may affect jaws after traumas, surgeries, previous infections, or in patients with other diseases. In the head and neck region, the condition appears as swelling associated with osteomyelitis and often with one or more draining, from the medullary spaces to skin or sinuses [63]. Only a few cases of ulcerations on the tongue associated with A. israelii have been reported in the literature [48]. This diagnosis has been excluded after the microbiological culture on the biopsied specimen.

Considering other malignancies, lymphomas can appear as rapidly growing asymptomatic ulcerated swellings: Hodgkin’s lymphomas are very rare in the oral cavity but can affect the neck district, whereas, for non-Hodgkin’s lymphomas, the head and neck region represents the second most common extra-nodal site [41,64]. Thus, a relapse of Acute Myeloid Leukemia or a new manifestation of any other hematological malignancy was excluded through pathology, which revealed an inflammatory process in association with purulent necrosis and microbial colonies. Considering other non-hematological malignancies, only verrucous carcinoma could be considered, but the rapid onset and the multifocal nature of these lesions immediately excluded such diagnosis; the following pathology confirmed the exclusion.

Finally, some orofacial granulomatosis should be considered in the differential diagnosis since most of them can present as oral ulcerated swellings. Wegener granulomatosis, Crohn’s disease, sarcoidosis, and Melkersson–Rosenthal Syndrome are the most relevant.

Wegener granulomatosis is a rare immune-based inflammatory necrotizing vasculitis of unknown cause, classically characterized by involvement of upper respiratory tract, lungs, and kidneys. The typical oral manifestation is *strawberry gingivitis*; less frequent findings include oral ulceration, necrosis, and perforation of the nasal septum or palate.

Crohn’s disease may affect the gastrointestinal tract from mouth to anus. A wide range of nonspecific oral lesions has been associated with this condition, including ulcers (aphthous-like, with a granulomatous appearing, or linear aspect), diffuse or nodular swellings, a cobblestone appearance of the mucosa, or macules and plaques involving the gingiva.

Sarcoidosis is a multisystem granulomatous disease of unknown cause that can affect any organ but the lungs, the lymph nodes, the skin, the eyes, and the salivary glands are the predominant sites. Oral lesions may occur on any surface, in most cases, as a submucosal mass, ulceration, a nodular swelling, an area of granularity, or an isolated papule. When this condition is characterized by recurring facial paralysis, swelling of the lips and a fissured tongue is called Melkersson–Rosenthal Syndrome.

Melkersson–Rosenthal Syndrome intraoral lesions can include edema, ulcers, papules, swellings, cobblestone mucosal alterations, or focal areas of submucosal enlargement.

For all these entities, pathology reveals the presence of classic non-caseating granulomas together with some peculiar clinical features (i.e., abdominal pain, diarrhea, rectal blood loss, facial nerve paralysis) [65] or laboratory tests which can help in differentiating each other (i.e., presence of proteinase-3 antineutrophil cytoplasm antibodies—PR3-ANCA, myeloperoxidase antineutrophil cytoplasm antibodies—MPO-ANCA, elevated serum angiotensin-converting enzyme (ACE) levels) [66]. All these diseases were excluded since no granulomas were detected in the oral biopsy.

## 6. Conclusions

*S. clavata* infection may cause fulminant fungaemia with multi-organ involvement in immunosuppressed patients with a 60.2% mortality rate: middle-age patients, male gender and Acute Myeloid Leukemia should be considered risk factors. Some reports highlighted that previous therapy with Cytarabine should be considered as a further risk factor. Our case is the first report of intraoral lesions as the first manifestation of *S. clavata* in an immunodeficient patient. Proliferative oral lesions in the immunosuppressed hematological subject are always challenging for the oral medicine doctor since the immune system response of these patients is abnormal: numerous benign and malignant diseases (i.e., lymphoma, other hematological malignancies, infectious diseases, granulomatous diseases) must be excluded. In this regard, microbiological sampling is fundamental in order to exclude rare mycotic infections.

## Figures and Tables

**Figure 1 ijerph-18-02385-f001:**
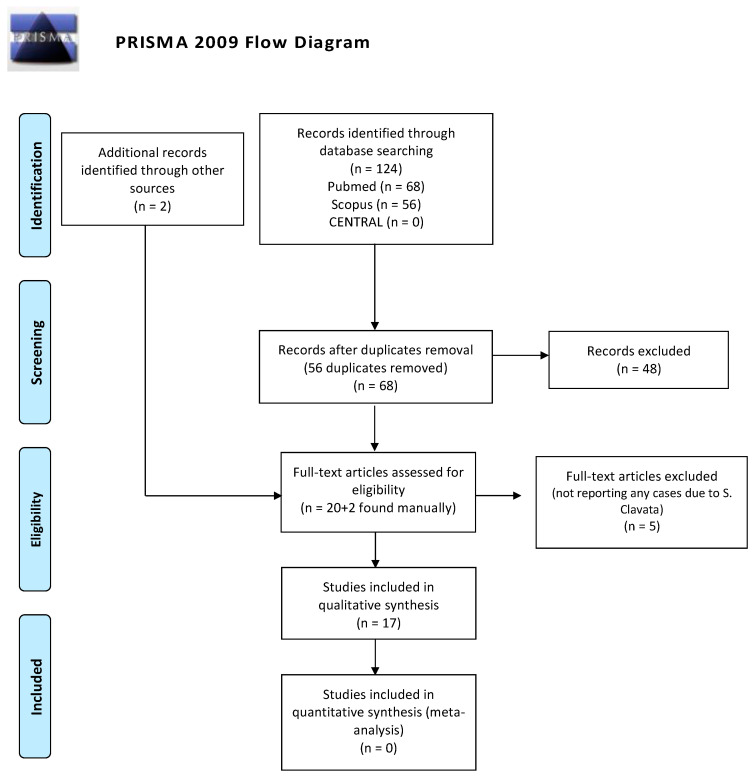
PRISMA (Preferred Reporting Items for Systematic Reviews and Meta-Analyses) flow-chart of the systematic review.

**Figure 2 ijerph-18-02385-f002:**
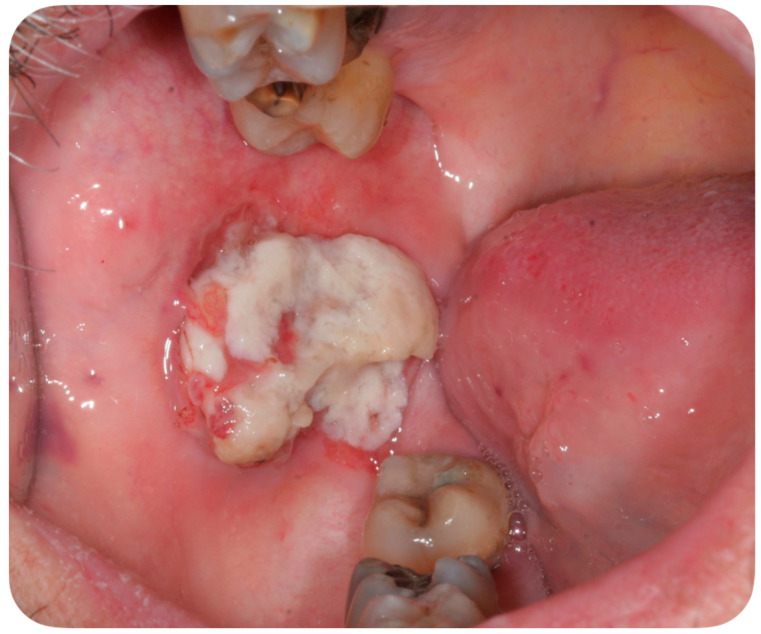
Asymptomatic ulcerated lesion, located on the right buccal mucosa.

**Figure 3 ijerph-18-02385-f003:**
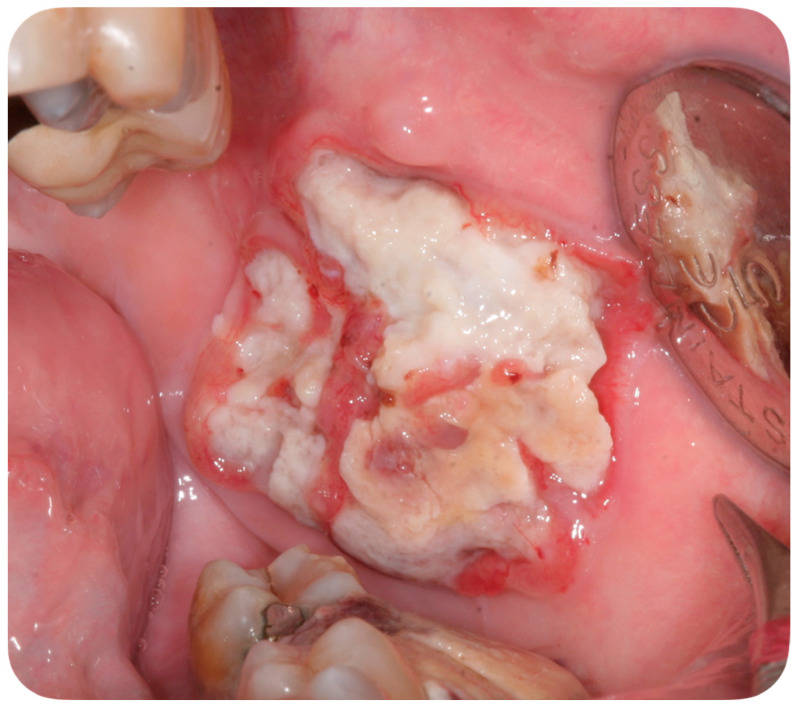
Asymptomatic ulcerated lesion, located on the left buccal mucosa.

**Figure 4 ijerph-18-02385-f004:**
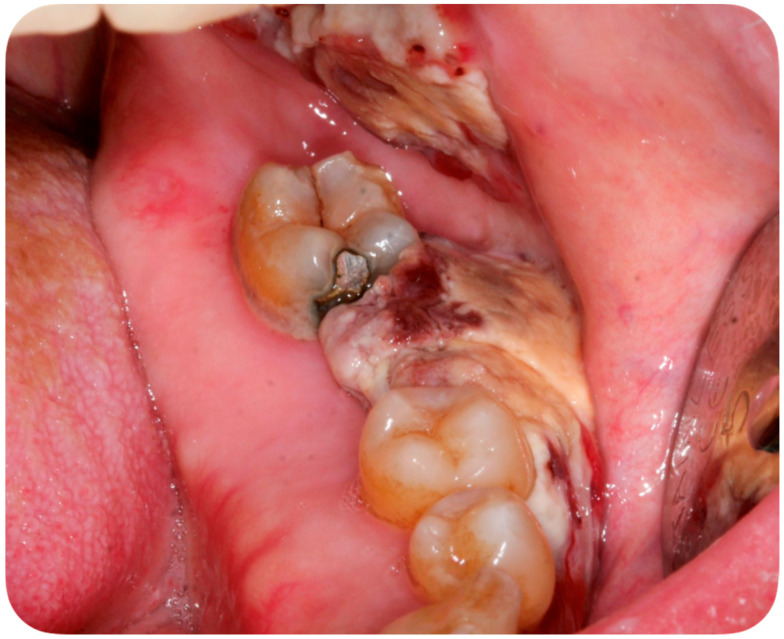
Asymptomatic ulcerated lesion, located on the left mandible.

**Table 1 ijerph-18-02385-t001:** Pooled data from a literature review, stratified according to geographic distribution, age, gender, anamnesis, and clinical features.

Study	N. of Cases	Country	Gender	Age	Underlying Disease	Clinical Form	Treatment	Outcome
**Lacroix et al., 2007** [15]	2	France	M	14	AML	Disseminated	VCZ, Amph	Survived
			M	59	AML	Disseminated	PCZ, Amph, 5-F	Survived
**Picard et al., 2014** [16]	3	France	F	46	AML	Disseminated	VCZ, Amph	Died
			M	70	AML	Disseminated, Pulmonary	Caspo	Died
			F	63	AML	Disseminated	VCZ, Amph	Died
**Vaux et al., 2014** [9]	30	France	M (15), F (15)	63 (mean)	AML (21), ALL (6), Other (3)	Disseminated (26), Pulmonary (12), Diarrhea (18)	Ns	Died (24), Survived (6)
**Camus et al., 2014** [14]	1	France	M	32	AML	Disseminated, Peritonitis, Hepatic Lesions	VCZ	Survived
**Del Principe et al., 2016** [4]	3	Italy	F	36	AML	Pulmonary, Cholecystitis, Hepatosplenic abscesses	VCZ, Amph	Survived
			F	50	Lymphoma	Disseminated, Pulmonary, Splenic infiltrated	VCZ, Amph	Died
			M	21	AML	Splenic Abscesses	VCZ, Amph	Survived
**Favre et al., 2016** [12]	1	France	M	27	AA	Disseminated	VCZ, Amph	Survived
**De Almeida et al., 2016** [17]	1	Brazil	F	6	Other	Ns	VCZ, Amph	Died
	18	Italy	M (11) F (6) Ns (1)	Ns	AML (8), Lymphoma (3), AA (2), Other (3), Ns (2)	Disseminated (18)		Ns

**Esposto et al., 2018** [18]							Ns	


**Liu et al., 2018** [19]	1	China	M	10	ALL	Disseminated, Pulmonary	VCZ, Mica, Amph	Survived
**Salguero-Fernandez et al., 2018** [20]	1	Spain	M	47	Lymphoma	Disseminated, Skin	Amph, 5-F	Died
**Leoni et al., 2018** [21]	1	Italy	M	6	Other	Disseminated, Pulmonary, Skin, Renal	VCZ, Amph	Survived
**Buchta et al., 2019** [22]	11	Czechia	M	45	AML	Disseminated	VCZ, Amph	Died
		Czechia	F	61	AML	Disseminated	Amph	Died
		Czechia	F	63	AML	Disseminated	VCZ, Amph	Survived
		Czechia	F	58	AML	Disseminated, Pulmonary	VCZ, Amph	Died
		Czechia	F	50	AML	Disseminated, Pulmonary	Amph	Died
		Czechia	F	66	Lymphoma	Disseminated	VCZ, Mica	Died
		Turkey *	F	37	AML	Disseminated	VCZ	Survived
		Israel *	F	17	AML	Disseminated, Liver, Spleen, CNS	VCZ, Amph, 5-F	Survived
		Spain *	M	48	Lymphoma	Disseminated, CNS, Liver, Pulmonary, Spleen	VCZ, Amph, 5-F	Died
		Germany *	M	55	AML	Disseminated	VCZ, Amph	Survived
		Serbia *	M	19	ALL	Disseminated, Pulmonary	Caspo	Died
**Pavone et al., 2019** [23]	1	Italy	F	54	PCKD	Disseminated, Peritonitis	VCZ, Amph	Died
**Wee et al., 2019** [24]	1	Singapore	M	13	ALL	Disseminated, Kidney, Liver, Skin	VCZ, Amph	Survived
**Stanzani et al., 2019** [10]	4	Italy	M	66	Lymphoma	Pulmonary	VCZ	Survived
			F	48	AML	Disseminated, Pulmonary, Liver, Spleen, Kidney	VCZ, Amph	Survived
			M	34	Lymphoma	Disseminated	VCZ	Survived
			M	64	AML	Disseminated, Pulmonary, Spleen, CNS	VCZ, Amph	Died
**Lo Cascio et al., 2020** [25]	7	Italy	M (6) F (1)	41.1 (mean)	AML (5), ALL (1), Lymphoma (1)	Disseminated (7), Diarrhea (3)	Amph (3), Echi (2), AZ (2)	Died (3) Survived (4)
**Menu et al., 2020** [26]	9	France	M (6) F (3)	57.8 (mean)	AML (4), Lymphoma (2), ALL (1), Other (2)	Disseminated (8), Pulmonary (3), Diarrhea (3) Digestive Symptoms (2)	VCZ (4), PCZ (3), Echi (2)	Died (5) Survived (4)
**This study**	1	Italy	M	56	AML	Oral Lesions	VCZ	Survived

AML: Acute Myeloid Leukemia; ALL: Acute Lymphoid Leukemia, AA: Aplastic Anemia, PCKD: Polycystic Kidney Disease, CNS: Central Nervous System, Ns: Non-specified, VCZ: Voriconazole, PCZ: Posaconazole, Mica: Micafungin, Caspo: Caspofungin, Amph: Amphotericin B; Echi: Echinocandins; AZ: Azoles; 5-F: 5-fluorocytosine. * Cases retrieved by the international registry FungiScope.

**Table 2 ijerph-18-02385-t002:** Main results of the review. * Mortality rate was calculated excluding the articles not reporting the final outcome of patients.

Age	
Years	51.8 (mean)
**Gender**	
Male	57
Female	38
Non-Specified	1
**Country**	**Tot 96**
France	46
Italy	35
Czech	6
Other Countries (Spain, Germany, Brazil, Turkey, Israel, Serbia, China, Singapore)	9
**Outcome**	**Total Number (*mortality rate* *)**
Death	47 (*60.2%*)
Survival	31 (*38.8%*)

**Table 3 ijerph-18-02385-t003:** Differential diagnosis for oral manifestation of *Saprochaete clavata.*

Clinical Entity	Etiology and Pathogenesis	Clinical Manifestation			Diagnosis	Reason for Exclusion during the Diagnostic Flow-Chart
		Usual Features	Unusual Oral Presentation	Site		
**Opportunistic Infections**						
***Candidiasis*** [34]	*C. Albicans and other C.* spp.	*Pseudomembranous Candidiasis:* white, soft plaques, removable, sometimes burning sensation and altered taste. *Acute erythematous candidiasis:* Erythema, usually painful	Rapidly growing exophytic lesions	Buccal mucosa Palate Dorsal tongue	Microbiological culture	Usually do not comprise rapidly growing asymptomatic ulcerated swellings
***Aspergillosis***	*Aspergillus* spp.	Oral manifestations do not arise in immunocompetent hosts	*Invasive form:* swelling, ulceration, necrosis, usually painful	Paranasal sinuses Oropharynx Palate Dorsal Tongue	Microbiological culture	(a) Uncommon location of the lesions (b) Absence of symptoms (c) Microbiological culture
***Histoplasmosis***	*H. Capsulatum*	Histoplasmosis of the head and neck is rarely seen in immunocompetent patients	Rapidly growing asymptomatic oral ulceration with firm margins, single or multiple, nonhealing.	Tongue, Palate, Buccal mucosa	Microbiological culture and pathology	(a) Oral Histoplasmosis usually does not appear as a friable swelling (b) Medical history: no travels in US (c) Microbiological culture and pathology
***Syphilis*** [35,36]	*T. Pallidum*	*Primary syphilis* indurated ulcer, asymptomatic. *Secondary syphilis:* red rash characterized by maculopapular areas, oral ulcers covered by membrane, or *condyloma lata*	Vascular proliferation, multiple ulcerated areas, hemorrhage	Lips, but any other site can be involved	Direct detection of *T. pallidum* and serologic testing (treponemal and non-treponemal tests, i.e., TPHA)	(a) Negative TPHA (b) Absence of characteristic systemic symptoms (secondary syphilis) (c) Pathology (absence of proliferative endarteritis and infiltration of plasma cells)
***Tuberculosis*** [37]	*Mycobacterium Tuberculosis*	Oral lesions are uncommon and occur due to infected sputum or hematogenous spread	Chronic and indurated ulcer, non-healing extraction sockets, osteomyelitis, and mandibular swellings.	Tongue and palate but any mucosal surface can be involved. Bone of maxilla or mandible	Molecular tests (nucleic acid amplification test—Xpert MTB/RIF), Pathology, microbiological tests (mycobacterial growth indicator tube—MGIT)	(a) Pathology (absence of typical caseous tuberculous granulomas) (b) Clinical presentation: Tuberculosis oral lesions usually have an “infiltrating” feature more than exophytic
***Actinomycosis*** [38,39]	*Actinomyces* spp.	Fibrosis, swellings, cutaneous draining sinus tracts	Ulcerations of the tongue and osteomyelitis. One reported case of involvement of floor of the mouth and buccal mucosa	Mandible and surrounding tissues. Salivary glands	Microbiological culture and pathology	Microbiological culture and pathology (absence of granulomatous inflammatory response)
**Malignant Diseases**						
***Lymphomas*** [40,41,42]	Heterogeneous malignant disease of the lymphatic system	*Hodgkin’s lymphomas:* very rare in the oral cavity. *Non-Hodgkin’s lymphomas*: rapidly growing asymptomatic ulcerated swellings, bone resorption, or bone loss	Pathologic fracture. Pain, numbness of the lip.	Tonsils, Salivary glands, maxilla, base of the tongue, Soft palate	Histopathological examination (presence of *Reed-Stemberg cells* for *Hodgkin’s lymphomas),* immunophenotyping, flow cytometry	Pathology and immunohistochemistry according to the lymphoma type
***Verrucous Carcinoma*** [43,44]	Exophytic variant of oral squamous cell carcinoma	Plaque-like or exophytic mass typically white, with a warty, ulcerated, or papillary surface. Slow continuous growth rate. Usually painless	Bone involvement (especially if invasive transformation occurs)	Buccal mucosa, Gingiva	Pathology (well-differentiated epithelial cells mass extending into the connective tissue generally with a pushing appearance)	(a) Pathology (b) Rapid onset (c) Multiple lesions
**Orofacial Granulomatosis**						
***Wegener Granulomatosis*** [45]	Rare immune-based inflammatory necrotizing vasculitis of unknown cause	*Strawberry gingivitis*, sinusitis, oral ulceration	Facial paralysis, Labial mucosal nodules, Necrosis and perforation of the nasal septum or palate, Swelling and desquamation of the lips, Salivary gland enlargement, Arthralgia of the TMJ Tongue involvement	Gingiva but any mucosal surface can be involved. Rarely major salivary glands	Pathology Clinical diagnostic criteria Presence of proteinase-3 antineutrophil cytoplasm antibodies (PR3-ANCA) myeloperoxidase antineutrophil cytoplasm antibodies (MPO-ANCA)	(a) Pathology (absence of granulomatous lesions, with necrotizing vasculitis) (b) Absence of clinical diagnostic criteria (Oral ulcerations or nasal discharge, nodules on chest radiograph, abnormal urinary sediment, granulomatous inflammation upon biopsy)
***Crohn’s Disease*** [46,47]	Chronic inflammatory disease of the gastrointestinal tract	Ulcers, diffuse or nodular swellings, cobblestone appearance of the mucosa, macules and plaques involving the gingiva	Angular cheilitis Alveolar bone loss	Lips Gingiva Buccal mucosa	Endoscopy Pathology	(a) Pathology (absence of classic *non-caseating granulomas*) (b) Absence of peculiar clinical features
***Sarcoidosis*** [48]	Multisystem granulomatous disease of unknown cause that can affect any organ	Oral manifestations are uncommon and associated with salivary gland and lymph node involvement	Submucosal mass, ulcerations, nodular swellings, an area of granularity, or an isolated papule. Bone involvement	Salivary glands Buccal mucosa but any mucosal surface can be involved	Symptoms Radiology Elevated serum angiotensin-converting enzyme (ACE) levelsPathology	(a) Pathology (absence of classic *non-caseating granulomas*) (b) Absence of peculiar clinical features
***Melkersson-Rosenthal Syndrome*** [49]	Rare disorder of unknown cause	Recurring facial paralysis, swelling of the lips, and a fissured tongue	Edema, ulcers, papules, swellings, cobblestone mucosal alterations, or focal areas of submucosal enlargement	Lips Tongue	Pathology Clinical diagnostic criteria	(a) Pathology (absence of classic *non-caseating granulomas*) (b) Absence of clinical diagnostic criteria (labial swelling, facial paralysis, and fissured tongue)

## Data Availability

Not Applicable.

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
