# Peer review of "Saprochaete clavata Infection in Immunosuppressed Patients: Systematic Review of Cases and Report of the First Oral Manifestation, Focusing on Differential Diagnosis"

_ijerph, 2021, doi:10.3390/ijerph18052385_

Round 1

Reviewer 1 Report

This paper is the study about “Saprochaete Clavata in Immunosuppressed Patients: Systematic Review of Cases and Report of Oral Lesions as first Manifestation of the Infection”. We think that this study is interesting. However, the concept and significance are unclear as systematic review. Please make the purpose of this study clear and consider reconsideration. Some comments are as below:

  1. If a systematic review is the purpose, a review of lesions in the oral cavity is needed.
  2. It's strange that a case report comes out.“”. There is no connection.

If so, citations are needed, and discussions only state the possibilities.

This is Inappropriate.

  1. If necessary, Figure4, please change to the appropriate oral photo.
  2. Do you need Table2?

Not required for reviews.

  1. A systematic review should essentially extract high quality (sample size/RCT) studies. This research cannot be said to be of high quality.

Author Response

This paper is the study about “Saprochaete Clavata in Immunosuppressed Patients: Systematic Review of Cases and Report of Oral Lesions as first Manifestation of the Infection”. We think that this study is interesting. However, the concept and significance are unclear as systematic review. Please make the purpose of this study clear and consider reconsideration. Some comments are as below:

We would like to thank the reviewer for the useful comments. We edited the title in order to better describe the article contents.

  1. If a systematic review is the purpose, a review of lesions in the oral cavity is needed.

We would like to thank the reviewer for the interesting comments: according to the literature review, this is the first case of oral manifestation and no other cases were retrieved. We discussed this in the introduction

n section, with special emphasis to the differential diagnosis of oral uncommon lesions. (lines 61-64)

  1. It's strange that a case report comes out.“”. There is no connection.

If so, citations are needed, and discussions only state the possibilities.

This is Inappropriate.

We apologize the reviewer, but the text in the “” was deleted and we couldn’t understand the suggestion.

If necessary, Figure4, please change to the appropriate oral photo.

Thank you for the comment, we have uploaded a higher quality image.

  1. Do you need Table2? Not required for reviews.

Thank you for the suggestion, we deleted the table and added the information in the text as suggested by reviewer 3. (lines 244-245)

  1. A systematic review should essentially extract high quality (sample size/RCT) studies. This research cannot be said to be of high quality.

We would like to thank the reviewer for the interesting comment. We strongly agree with the reviewer that systematic review should only include high quality articles. Unfortunately, being the S. clavata an emerging pathogen, the available literature only provides case reports and case series. Nevertheless, we thought that the topic could be interesting even including lower quality standard studies, hoping that scientifically sounder papers will be published in the future, as we stated into the discussion section. (line 198-207)

.

Submission Date

13 January 2021

Date of this review

16 Jan 2021 04:23:21

Reviewer 2 Report

The article is a very interesting case report and literature review on  this very rare fungal infection caused by Saprochaete Clavata.

The article adds new information to the literature, as it is the first review on this particular argument. 

Only minor comments:

  • I would probably add in table 1 also the tretment proposed for each infection and the dosage.
  • - I would probably insert another table summarizing the main derived data from the review (mortality rate, etc.....) 
  • In matherial and methods section, I would add the statistical program that allowed you to do all the analysis                

Reviewer 3 Report

The Authors with this systematic review entitled “Saprochaete Clavata in immunosuppressed patients: systematic review of cases and report of oral lesions as first manifestation of the infection” aimed to report a literature revision of all cases of Saprochaete Clavata (SC) infection and to describe the first oral lesion as manifestation of SC infection.

In my opinion the main relievable problem of this manuscript is due to the un-linkage between revision of the literature and reported case report. I suggest to the Authors to present the case as an example of the data obtained by the analysis of the literature about SC. The two part should be more interconnected and case report should be integrated in the discussion with the data obtained from the literature.

Data obtained from the literature as reported in table 1, are not sufficiently augmented in discussion section. Discussion considered mainly differential diagnosis to define oral lesions but only marginally about SC lesion.

The manuscript is written in fluent English without typing errors expect at the line 76 where a double semicolon was typed.

In the results section:

Data cited in the text (from line 86 onwards) about the number of analysed articles are not in line with data reported in the diagram of the figure 1. For example, it is not clear how the Authors have selected 22 articles starting from 114, plus 2 (found manually), and then the exclusion of 58 articles and removal of 46. Please review the flow chart reformulating the disposition of the logical sequence.

Figure 1 is cited in line 91, after reporting the number of patients, but in figure 1 the number of patients is not reported. The phrase seems to be related to Table 1.

About the case report, if possible, the photos showing the “resolution of the oral lesion” could be interesting.

Author Response

The Authors with this systematic review entitled “Saprochaete Clavata in immunosuppressed patients: systematic review of cases and report of oral lesions as first manifestation of the infection” aimed to report a literature revision of all cases of Saprochaete Clavata (SC) infection and to describe the first oral lesion as manifestation of SC infection.

In my opinion the main relievable problem of this manuscript is due to the un-linkage between revision of the literature and reported case report. I suggest to the Authors to present the case as an example of the data obtained by the analysis of the literature about SC. The two part should be more interconnected and case report should be integrated in the discussion with the data obtained from the literature.

Thank you. In the introduction section we clarified that the objective of the study is to review the reported cases, but even to discuss the differential diagnosis of rapidly growing ulcerative oral lesions. Nevertheless, we augmented the discussion with the analysis of systematic review data. (line 198-207/ 217-220/ 233-240)

Data obtained from the literature as reported in table 1, are not sufficiently augmented in discussion section.

Thank you for the useful comment. We augmented this analysis in the discussion section. (line 198-207/ 217-220/ 233-240)

Discussion considered mainly differential diagnosis to define oral lesions but only marginally about SC lesion.

Thank you for your comment. We tried to describe the lesion as accurately as possible. Since it is the first Oral manifestation caused by S. clavata, we thought to describe the diagnostic flow-chart which helped us to exclude the other possible diagnoses.

The manuscript is written in fluent English without typing errors expect at the line 76 where a double semicolon was typed.

Thank you, we edited the error.

In the results section:

Data cited in the text (from line 86 onwards) about the number of analysed articles are not in line with data reported in the diagram of the figure 1. For example, it is not clear how the Authors have selected 22 articles starting from 114, plus 2 (found manually), and then the exclusion of 58 articles and removal of 46. Please review the flow chart reformulating the disposition of the logical sequence.

Thank you for the useful comment. There was a mistake in the figure, we corrected it and made the flow chart clearer.

Figure 1 is cited in line 91, after reporting the number of patients, but in figure 1 the number of patients is not reported. The phrase seems to be related to Table 1.

Thank you, we corrected it.

About the case report, if possible, the photos showing the “resolution of the oral lesion” could be interesting.

We agree with the reviewer. Unfortunately, the patient was dismissed from the hospital before we could take the photos of the healed lesions. We planned a visit in our department, but the worsened conditions of the patient did not allow him to attend the scheduled visit.

Submission Date

13 January 2021

Date of this review

30 Jan 2021 14:23:34

Reviewer 4 Report

Saprochaete Clavata in Immunosuppressed Patients : Systematic Review of Cases and Report of Oral Lesions as first Manifestation of the Infection

Authors present an interesting review of invasive infection due to the rare yeast-like species, Saprochaete clavata. They also present a useful table about diagnostic.

Some minor revisions need to be done and especially concerning the name of the fungi. I also proposed some other modifications and some details should be added about methods used in the case report.

For the name of the species there is only capital letter for the genus, and name of species are Latin and should be in italic.

Line 39 why the term « genus » is in capital letter ?

Line 40 , Do you mean : Geotrichum can be considered as an emerging pathogen or agents ?

Line 50 please change the reference because reference 9 (Vaux et al,) doesn’t indicate the S. clavata or S. capitata are used for cheese maturation, normally the species frequently used for cheese industry in Geotrichum candidum. Invasive infection due to S. clavata seem to have oral origine but in the studies cited, dairy products and dishwasher seem to be the main origine.

Table 1 why the study presented by Menu et al (reference 17) is not in this table ? Thus study present 9 cases of infection due to S. clavata.

Line 154 which method for the identification of the species ?

Table 2 : some additional information are needed (measure of MIC (mg/L), method used for MIC determination …)

Line 177 « the » before S. clavata is not useful

Line 190 In the case report, could the authors explain how they can conclude that yeast recovered from mouth are the first site of infection ? COuld you please indicate if any bloodculture or any other samples were performed before for this patient ? any skin lesions ?

Line 206 culture instaed of colture

Line 211 Aspergillosis spores colonize also mainly lung

Round 2

Reviewer 3 Report

Dear Authors,

Thank you for the effectuated revisions, but I think that there are some points to modified.

From line 101 to line 102 the font is different from the text. Main concerns are relative to lines from 104 to 110 and relative table 1.

In line 105 the reported cases with gastrointestinal (diarrhea) are 24 but in the table the count is 21, about pulmonary disease are 25, but in the table the count is 27, about Lymphomas are 10 but in the table the count is 12 and about ALL reported are 8 vs 10 cases in the table.

In the caption of the table 1, please add “h” after Amp as reported in the table 1. Always in the table 1, I suggest changing “lung” with “pulmonary”, in order to help the reading. Further in the text in line 108 is cited “PCKD” that is not detected in the table.

Best Regards

Author Response

Dear Authors,

Thank you for the effectuated revisions, but I think that there are some points to modified.

From line 101 to line 102 the font is different from the text. Main concerns are relative to lines from 104 to 110 and relative table 1.

In line 105 the reported cases with gastrointestinal (diarrhea) are 24 but in the table the count is 21, about pulmonary disease are 25, but in the table the count is 27, about Lymphomas are 10 but in the table the count is 12 and about ALL reported are 8 vs 10 cases in the table.

Thank you for your comments, we corrected our mistakes.

In the caption of the table 1, please add “h” after Amp as reported in the table 1. Always in the table 1, I suggest changing “lung” with “pulmonary”, in order to help the reading. Further in the text in line 108 is cited “PCKD” that is not detected in the table.

Thank you for your comments, we edited the table according to your useful suggestions.

Best Regards